# The Neurological and Hemodynamics Safety of an Airway Clearance Technique in Patients with Acute Brain Injury: An Analysis of Intracranial Pressure Pulse Morphology Using a Non-Invasive Sensor

**DOI:** 10.3390/s24217066

**Published:** 2024-11-02

**Authors:** Daniela de Almeida Souza, Gisele Francini Devetak, Marina Wolff Branco, Reinaldo Luz Melo, Jean Lucas Tonial, Ana Marcia Delattre, Silvia Regina Valderramas

**Affiliations:** 1Postgraduate Program in Internal Medicine and Health Sciences, Empresa Brasileira de Serviços Hospitalares, Universidade Federal do Paraná, Curitiba 80060-900, Brazil; 2Empresa Brasileira de Serviços Hospitalares, Universidade Federal do Paraná, Curitiba 80060-900, Brazil; gisele.casarotti@ebserh.gov.br; 3Postgraduate Program in Internal Medicine and Health Sciences, Universidade Federal do Paraná, Curitiba 80060-000, Brazil; 4Department of Medicine, Universidade Federal do Paraná, Curitiba 80060-000, Brazil; 5Department of Prevention and Rehabilitation in Physiotherapy, Universidade Federal do Paraná, Curitiba 80060-000, Brazil

**Keywords:** mechanical ventilation, mucociliary clearance, ventilator hyperinflation, ICP wave morphology, intracranial compliance, intracranial pressure, non-invasive intracranial pressure monitor

## Abstract

Patients with acute brain injury (ACI) often require mechanical ventilation (MV) and are subject to pulmonary complications, thus justifying the use of Airway Clearance Techniques (ACTs), but their effects on intracranial pressure (ICP) are unknown. This study investigates the neurological and hemodynamics safety of an ACT called ventilator hyperinflation (VHI) in patients with ACI. This was a randomized clinical equivalence trial, which included patients aged ≥ 18 years with a clinical diagnosis of hemorrhagic stroke, with symptom onset within 48 h. The participants were randomly allocated to the Experimental Group (EG, *n* = 15), which underwent VHI followed by tracheal aspiration (TA), and the Control Group (CG, *n* = 15), which underwent TA only. Neurological safety was verified by analyzing the morphology of the ICP wave through the non-invasive B4C sensor, which detects bone deformation of the skull, resulting in a P2/P1 ratio and TTP, and hemodynamics through a multi-parameter monitor. Evaluations were recorded during five instances: T1 (baseline/pre-VHI), T2 (post-VHI and before TA), T3 (post-TA), T4 and T5 (monitoring 10 and 20 min after T3). The comparison between groups showed that there was no effect of the technique on the neurological variables with a mean P2/P1 ratio [F (4,112) = 1.871; *p* = 0.120; np2 = 0.063] and TTP [F (4,112) = 2.252; *p* = 0.068; np2 = 0.074], and for hemodynamics, heart rate [F (4,112) = 1.920; *p* = 0.112; np2 = 0.064] and mean arterial pressure [F(2.73, 76.57) = 0.799; *p* = 0.488; np2 = 0.028]. Our results showed that VHI did not pose a neurological or hemodynamics risk in neurocritical patients after ACI.

## 1. Introduction

Stroke is among the leading causes of death and disability worldwide [1]. Due to its complexity, acute brain injury (ABI) often involves intracranial and extracranial complications, which can cause secondary brain damage and worsening of clinical condition [1]. The use of mechanical ventilation (MV) for this patient profile is common and involves numerous factors, such as neurological protection and, more recently, pulmonary protective ventilation [2].

The relationship between ventilatory adjustments and cerebral hemodynamics in ACI is still the subject of constant discussions in the literature [3,4,5]. The dynamics between the increase in intrathoracic pressure (ITP) resulting from positive pressure and its influence on venous return and cerebral perfusion pressure (CPP) is one of the central points in ventilatory adjustment in critically ill neurological patients [4]. Also relevant are the effects of arterial partial pressure of carbon dioxide (PaCO_2_) on cerebral autoregulation, which can have vasodilator or vasoconstrictor effects, depending on narrow changes in the normal range, with the former still being responsible for increasing ICP [2].

Neurocritical patients on MV have multifactorial causes for the accumulation of secretions in their airways, such as the presence of an orotracheal tube, decreased cough reflex and effectiveness, rheological changes in mucus, immobility and consequent muscle weakness, longer MV time and slower weaning in comparison to other intensive care unit (ICU) populations, and immunosuppression induced by the ischemic event [6,7,8,9,10]. Secretion retention in the airways is related to a higher incidence of pneumonia and atelectasis, which increase hypoxia, MV time and ICU stay, as well as increased morbidity and mortality rates [11,12,13].

Among the Airway Clearance Techniques (ACT), Ventilation Hyperinflation (VHI) has gained ground in recent years due to its benefits over other techniques, such as control of the parameters utilized with well-established safety limits and the possibility of the bedside assessment of its effectiveness on ventilatory mechanics, its lower risk of circuit contamination, cyclical opening and closing injury and loss of lung recruitment [14,15]. Recent studies have concluded that despite the potential benefits of VHI regarding ventilatory mechanics and the removal of pulmonary secretions, there is still limited data preventing its widespread use in clinical practice [14,15]. Among these impediments, it is important to highlight the neurological safety of applying the technique, since the influence of intrathoracic positive pressure on cerebral hemodynamics is still a constant concern in clinical practice, with conflicting data [10,16,17].

The gold standard for neuromonitoring is the use of transducers connected to extraventricular drainage; however, they may not be indicated in certain clinical conditions due to the risk of complications, such as direct injury to brain tissue, hemorrhages and infections [18,19,20]. In this context, non-invasive monitoring techniques have been developed, amongst which the skull microdynamics sensor (B4C) stands out, which evaluates the mechanical properties of the brain [21]. Developed by a Brazilian engineer who hypothesized a new understanding of the Monro–Kellie doctrine, the B4C has been validated to monitor intracranial pressure (ICP) and intracranial compliance (ICC) [22,23,24]. This sensor allows the morphology of the ICP wave to be analyzed, which can shed light on cerebral pathophysiology [4]. Although waveform analysis is a well-known parameter, its use in clinical practice is limited as invasive systems often do not analyze it in an automated way, which makes its interpretation subjective [21,25]. ICC is defined as the ability to accommodate volume alterations without an increase in pressure, i.e., keeping the intracranial content in balance [26].

Thus, considering the feasibility of neuromonitoring to dynamically analyze the effects of increased ITP over the ICP, as well as the use and need for ACT in neurocritical patients on MV, the aim of this study was to assess the effects of ACT in the VHI modality on cerebral (from the point of view of ICP wave morphology) and systemic hemodynamics, seeking evidence to support clinical practice in this population.

## 2. Materials and Methods

This randomized controlled clinical trial was approved by the Research Ethics Committee of the Complexo Hospitalar de Clínicas of the Federal University of Paraná (CAAE: 01378118.7.0000.0096) and registered on the Brazilian Clinical Trials Registry Platform (RBR-5qs9k2n). The protocol was designed in accordance with the Consolidated Standards of Reporting Trials—SPIRIT.

This study included patients of both sexes, over 18 years of age, admitted to the ICU with a hemorrhagic stroke diagnosis confirmed by neuroimaging, with an onset of neurological symptoms within 48 h, with invasive ventilatory support via an intubated orotracheal tube after the neurological event, without a previous diagnosis of pulmonary or neurological disease. Patients with a Richmond Agitation–Sedation Scale (RASS) score greater than −3 (difficulty calculating pulmonary mechanics, asynchrony in MV and difficulty coupling the B4C sensor in agitated patients), hemodynamics instability (increase in vasoactive amines or lactate in the last 24 h), decompressive craniectomy or presence of open external ventricular drainage (EVD) were excluded. All family members signed an informed term of consent form before the subjects were included in this study. The subjects underwent simple randomization, and the allocation was concealed using opaque, sequentially numbered and sealed envelopes. All procedures were carried out by an independent researcher. Participants were allocated to the Control Group (CG), receiving tracheal aspiration alone (TA), and the Experimental Group (EG), undergoing VHI and TA. Figure 1 shows the flowchart of recruitment, interventions and participant assessments.

Two hours before the collections began, the patients were positioned in a supine position, at 30°, and underwent TA and adjustment of the ventilatory parameters (pre-intervention parameters). The team was instructed not to disconnect the patient from the MV until the collections were completed.

The assessments were carried out at five different times: T1, T2, T3, T4 and T5. Each assessment consisted of measuring neurological parameters (ICP morphology using the B4C sensor), hemodynamics parameters (heart rate (HR), mean arterial pressure (MAP) and peripheral oxygen saturation (SpO_2_)) and pulmonary mechanics (drive pressure, plateau pressure and peak pressure). Figure 2 shows the schematic design of the collection period. Details of the study protocol can be analyzed in Ref. [27].

The morphology of the ICP wave was measured using the B4C sensor (Model BcSs-PICNI2000), which detects and monitors cranial bone deformations through the micrometric pulsation of this pressure, which changes according to the intracranial volume–pressure curve [21,24,28]. The propagation of the arterial pulse at the level of the choroid plexus into the cerebrospinal fluid and the brain parenchyma generates the ICP pulse waveform, which has three components: P1, P2 and P3 [21]. The P1 component corresponds to the transmission of arterial pressure from the choroid plexus to the ventricles (percussion wave), P2 to the ICC (tidal wave) and P3 to the closure of the aortic valve during diastole (dicrotic wave) [29]. Under normal conditions, the amplitudes of these peaks are P1 > P2 > P3, and the time elapsed (delay) from the start of the pulse waveform to the highest peak is short. This time is called the latency, rise time or time to peak (TTP) [29]. The change in the ratio between these peaks, especially if P2 > P1, is a marker of reduced ICC or ICH [30]. Figure 3 shows an example of the ICP curve obtained from the B4C sensor report.

The sensor was positioned on the scalp, without the need for any preparation (trichotomy, surgical incision or trepanation), in the frontotemporal region, approximately 3 cm above the first third of the orbitomeatal line, supported by a headband with a gentle clamping mechanism, by direct pressure contact with the scalp using a pin. Contact with the main branches of the superficial temporal artery and temporal muscle was avoided. The sensor is capable of detecting micrometric deformations of the cranial bone with sensitivity to cranial movements smaller than 0.2 μm [21,24]. The patients were monitored for 35 min, after T1, and all recordings were digitized, filtered and amplified, and the artifacts of the device were excluded, then transferred to a computer for posterior analysis.

The B4C system generates a report that includes parameters such as the P2:P1 ratio and TTP. The P2:P1 ratio comes from the division of the amplitudes of the P2 and P1 peaks, reflecting an increase in ICP at critical levels or the exhaustion of the compensatory reserve mechanisms, since the P2 peak increases its amplitude in relation to P1 [21,29]. The TTP, on the other hand, is calculated using the moment in time when the slope of the pulse is steepest up to the highest pulse (peak) [29]. Details of the physical principles of the B4C sensor can be found in the study by the authors of [29]. The P2:P1 ratio and TTP were calculated from the average of the pulses within each minute of monitoring in each evaluation time interval (T1 to T5). The reference parameters for the P2:P1 ratio and TTP were set at 0.6 to 0.8—very low risk of ICH, inactive compensation systems; >0.8 to 1.2*—low risk of ICH, active compensation systems; >1.2* to 1.4—potential risk of ICH, exhausted compensation systems; and >1.4 or TTP > 0.3 ms high risk of ICH, exhausted compensation systems (* for women ≥ 40 years, consider normality cutoff up to 1.4) [23,31,32,33].

Hemodynamics were assessed using a multi-parameter monitor (Mindray iMEC12) (Hamburg, Germany)and capnography using a Dräger Vamos^®^ monitor (Lübeck, Germany) and a sensor. Safety during the intervention was guaranteed by hemodynamics monitoring and maintaining the peak and plateau pressures below 40 and 30 cmH20, respectively. Pulmonary mechanics were measured using the flow interruption method at the end of inspiration with a three-second pause. Three measurements were taken using the mean of the measurements. In the absence of a plateau, the measurement was disregarded, either due to the presence of leaks or patient interference. No other ACT was associated with VHI.

The TA procedure followed the recommendations of the American Association for Respiratory Care [21] (2010), which establishes that patients should be ventilated with 100% FIO_2_ for 30 s before and 60 s after the procedure, reducing the risk of hypoxemia. Each patient underwent TA for 15 s, three consecutive times, using a closed suction system with a number 14 catheter.

Statistical analysis was carried out using JASP software (v. 0.18.1). The confidence interval (CI) was set at 95%, with a value of α ≤ 0.05 for every test applied. The normality of data distribution was tested using the Shapiro–Wilk test. Considering the parametric distribution observed in the quantitative variables, the results were presented as means ± standard deviations. The categorical variables were presented as absolute numbers and/or percentages of the total.

Demographic, clinical and anthropometric characteristics were compared between the groups (CG vs. EG) using the *t*-test for independent samples, after confirming the homogeneity of variances (Levene’s test) when the variables were quantitative, and Fischer’s Exact Test or the Chi-square test were used for categorical variables.

The comparison between CG and EG, considering the five monitoring moments (T1 to T5) was carried out using the two-way ANOVA test. Validations of sphericity (Mauchly test) were calculated, and corrections (Greenhouse–Geisser) were applied if necessary. Based on these validations, we applied Tukey’s post hoc analysis. The effect size was calculated by partial Eta (np2), whose adopted classification was based on Cohen (1988), where np2 values are considered small (<0.06), moderate (>0.06–0.14) and large (>0.14).

The sample calculation was carried out using G* Power 3.1^®^ statistical software based on the study by Olson et al. (2009), where a sample of 15 patients in each study group would provide 80% power (Type I error −α = 0.05, Type II error −β = 0.10).

## 3. Results

### Baseline Characteristics of the Patients

The sample consisted of 30 participants equally divided into the CG (*n* = 15) and the EG (*n* = 15), with intracerebral hemorrhage (76.66%) or subarachnoid hemorrhage (23.33%) assessed between 12 and 24 h since the beginning of the neurological event in 56.66% of the sample and between 24 and 48 h in 43.33%. The characterization of the sample, injury profile and comorbidities are described in Table 1.

During the application of BSRT in the EG, there was a 150% increase in VT, which consequently led to a significant increase in VTE, DP and Cst. Hyperventilation, in turn, significantly reduced ETCO_2_. Figure 4 shows the maximum range of VHI.

## 4. Neurological Safety of VHI

The results of the ANOVA comparing the two groups at the five monitoring times (T1, T2, T3, T4 and T5) have shown that there was no effect of the technique/method on the mean P2/P1 [F (4,112) = 1.871; *p* = 0.120; np2 = 0.063] and TTP [F (4,112) = 2.252; *p* = 0.068; np2 = 0.074], demonstrating the neurological safety of the VHI in the neurocritical patients in this study. Figure 5 shows the distribution of the P2/P1 and TTP data, respectively.

## 5. Hemodynamics Safety of VHI

ANOVA also revealed that there was no effect of technique/method on the mean values of the HR [F (4,112) = 1.920; *p* = 0.112; np2 = 0.064] and MAP [F(2.73, 76.57) = 0.799; *p* = 0.488; np2 = 0.028]. The mean values (SD) of the two groups referring to both HR and MAP are shown in Figure 6A,B. However, as can be seen in Figure 6B, there was a significant difference for both groups (*p* = 0.002), with an increase in mean MAP values immediately after aspiration (i.e., peak at T3) and a subsequent reduction in these values at T4 and T5.

With regard to the mean SpO_2_ values, ANOVA revealed a significant effect between the groups [F (2.53, 70.87) = 7.179; *p* < 0.001; np2 = 0.836], and post hoc analysis has shown an increase in mean SPO_2_ values (EG > CG) at T2, followed by a significant drop in both groups at T3 and consequent gradual recovery over time (T4 and T5). Details of the changes in SPO_2_ are shown in Figure 6C.

## 6. Discussion

The main finding of this study was that neurocritical ACI patients did not suffer significant changes in the morphology of the ICP wave assessed using a non-invasive sensor, following the application of a VHI-based ACT. According to an extensive literature search, this is the first clinical trial to carry out this research and be published. According to the study by Brasil et al. (2021), significant increases in the P2/P1 ratio verified by a non-invasive sensor are related to an increase in invasively measured ICP. The same author also recently stated that the P2/P1 ratio is recognized as a marker of impaired ICC and can be used as a valuable indicator of personalized ICP decompensation [21]. Studies show that ICC seems to be more relevant than ICP itself, as it reflects cerebral hemodynamics in a more individualized way. Thus, the brain seems to be tolerant of different ICP values, with different responses within numerical thresholds [28,34]. Tomar et al. (2019) compared the effects on ICP of two BTRS (manual thoracic percussion and percussion by mechanical vibrator) in 26 patients after severe traumatic brain injury (TBI) and concluded that only the manual technique showed statistically significant effects on ICP. The study by Cerqueira Neto et al. (2013) applied the expiratory flow acceleration technique to the same population and concluded that it did not influence ICP.

This trial included a sample of subjects with acute brain injuries from hemorrhagic strokes and found that this population had an altered ICC with active compensation systems, with a P2/P1 ratio of 1.16 (±0.176) and a TTP of 0.21 (±0.06) ms. This corroborates the data of De Moraes et al. (2022), who assessed 18 patients after ischemic and hemorrhagic strokes and observed a mean (SD) P2/P1 ratio and TTP of 1.17 (±0.326) and 0.219 (±0.104) ms, respectively. The authors correlated the analyses of invasive measurements by ventricular catheter with the B4C sensor and have shown a high association between the methods, since all patients with intracranial hypertension (mean ICP above 20 mmHg for at least 5 min) had a P2/P1 ratio greater than 1 and a TTP greater than 0.2. Although the baseline value shows that subjects with stroke have a change in the P2/P1 ratio and TTP, the brain still seems to be tolerant to increased ITP, without significant changes in ICP.

The increase in VT during the 10 min of VHI caused a drop in ETCO_2_ from 37 (±2.5) to 30.66 (±3.62) mmHg. This hypocapnia was not correlated with significant changes in the P2/P1 ratio and TTP, despite the reduction in both measurements. Hypocapnia is an expected consequence within VHI and is already an established conduct in the literature for the management of ICH. However, recent recommendations warn of its effects on cerebral oxygenation, since physiologically it reduces the cerebrovascular diameter and thus the intracranial volume, which is why it reduces ICP, but it can also reduce blood flow, causing possible cerebral ischemia [1,2,35,36]. A recent study involved hyperventilated patients with acute brain injury and ICH in a short time interval, with a range of mild hypocapnia (32–35 mmHg), and found a significant correlation with a reduction in ICP [37]. The literature is scarce in terms of data investigating the interaction between ICP, cerebral autoregulation and different CO_2_ levels [2].

The hemodynamics impact regarding respiratory physiotherapy in the ICU has been assessed using HR, MAP and SpO_2_ [38,39]. Our data have shown that there was no statistical difference in relation to systemic hemodynamics (HR and MAP) for the groups in relation to the method/time, which was also concluded by a recent systemic review with meta-analysis dealing with studies that utilized the same ACT, but with a cohort composed of a varied population, i.e., not just neurocritical patients [15]. Among the studies included in this review, which deal with the stability of systemic hemodynamics, VT increases ranged from 50% to 150%, maintained for a total maneuver time of 10 and 5 min, respectively [38,40].

In this study, the volume increase was 150% over 10 min, which guarantees the possibility of a longer maneuver time without impairing systemic hemodynamics. Although there was no statistical difference, the data analysis allows us to verify that, as expected, the increase in VT achieved by the maneuver, i.e., the increase in ITP caused by this increase in volume, led to a reduction in MAP in the EG, followed by an increase in MAP after aspiration in both groups. Thus, in a hemodynamically unstable population, the indication for this procedure may need to be reviewed or monitored more closely. The increase in MAP after aspiration has already been widely described in the literature and is probably related to sympathetic stimulation and consequent peripheral vasoconstriction [41,42].

Oxygenation as measured by SpO_2_ showed a statistical difference between the groups, which was maintained throughout the assessment time, which is possibly related to the effectiveness of the technique in promoting bronchial hygiene and lung expansion. This recruitment has also been pointed out by other studies as a pulmonary response to the technique [43,44]. It is also relevant that this increase in SpO_2_ prior to TA allowed the SpO_2_ in the EG not to fall below 90%, which guarantees greater safety for bronchial hygiene therapy in neurocritical patients. This recruitment-promoting characteristic is extremely relevant in an environment where lung collapse, and its consequences, such as a higher risk of infection and slower weaning, can increase the length of hospital stay and dependence on MV [12,13]. To our knowledge, there are no other studies that have assessed the systemic and neurological hemodynamic safety of BRST in a population with acute neurological injury, which has also been pointed out by other studies as a necessity for current clinical practice [45].

We can suggest that, despite the hypothesis that the increase in ITP could increase ICP as a measure of maintaining CPP after the fall in MAP, hyperventilation and its reduction in CO_2_ probably meant that VHI had no clinically relevant effects on cerebral hemodynamics.

The limitations of this study include that we did not blind the therapist and the fact that arterial CO_2_ pressure was not analyzed using arterial gasometry. Hemodynamically unstable patients were not included in this study, as we believe that subjecting them to a technique that tends to reduce MAP could lead to a reduction in CPP and secondary brain damage. Similarly, we did not include neurocritical patients with lung damage because the literature recommends that this patient profile should be ventilated individually, which would require a different protocol from the one designed for patients without lung damage. It is also important to mention that we only assessed the neurological safety of the technique by analyzing the morphology of the intracranial pressure wave, utilizing a sensor validated by the gold standard, although other neurological parameters could contribute to a more detailed neurological assessment. Therefore, further studies are needed that address a more complete neurological assessment, as well as the relationship between different ETCO_2_ values and parameters related to ICC.

## 7. Conclusions

In this way, despite the evident reduction in ICC in post-stroke patients, performing ACT that alters ITP does not seem to have a significant influence on ICP, and can be indicated for this patient profile as necessary. The role of non-invasive technologies in the bedside management of neurocritical patients is also relevant, making it possible to monitor and evaluate the effects of essential therapies to reduce morbidities related to hospitalization.

## Figures and Tables

**Figure 1 sensors-24-07066-f001:**
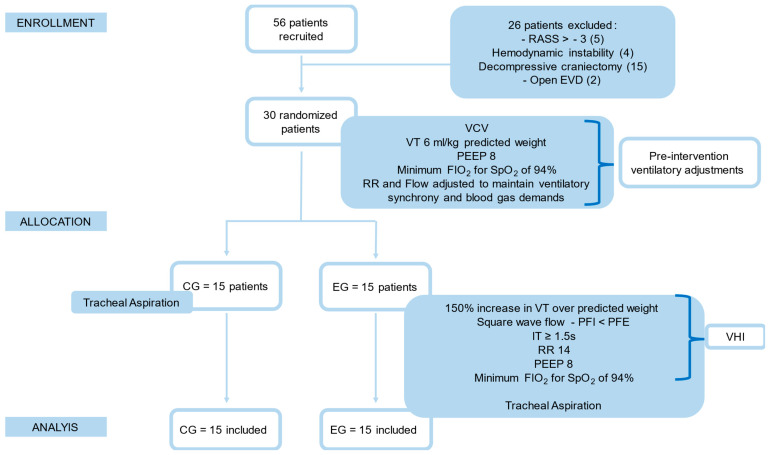
Flowchart of recruitment, interventions and assessments of participants. EG: Experimental Group; CG: Control Group; RASS: Richmond Agitation–Sedation Scale; EVD: External Ventricular Shunt; PFI: Peak Inspiratory Flow; PEF: Peak Expiratory Flow; PEEP: Positive End-Expiratory Pressure; FIO_2_: Fraction of Inspired Oxygen; SpO_2_: peripheral oxygen saturation; VT: Tidal Volume; F: Flow; RR: Respiratory Rate; IT: Inspiratory Time.

**Figure 2 sensors-24-07066-f002:**
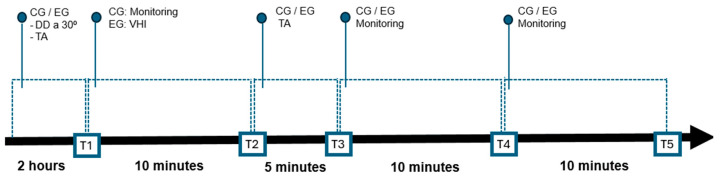
Schematic drawing of the experimental design. T1 to T5 represent evaluation moments. EG: Experimental Group; HMVM: Hyperinflation Mechanical Ventilation Maneuver; CG: Control Group; TA: tracheal aspiration; DD: Dorsal Decubitus.

**Figure 3 sensors-24-07066-f003:**
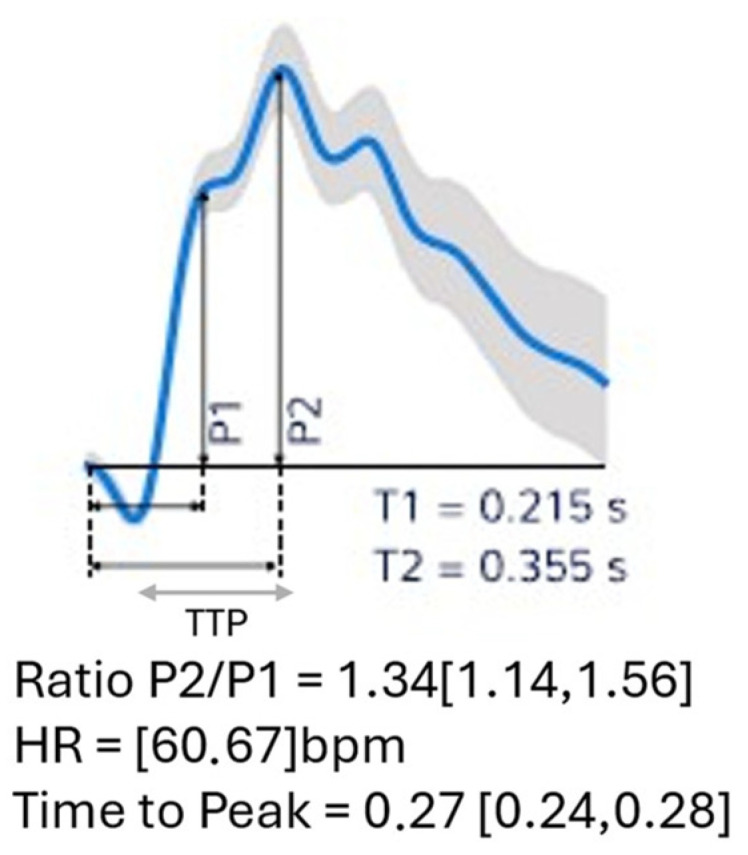
ICP curve obtained by the B4C sensor in a report. P2/P1 ratio: the ratio between the amplitudes of peaks P2 and P1; TTP: time to peak, defined as the time, from the start of the pulse, at which the ICP waveform reaches its highest peak. The shaded line represents the confidence interval of the 1 min monitoring. HR: heart rate.

**Figure 4 sensors-24-07066-f004:**
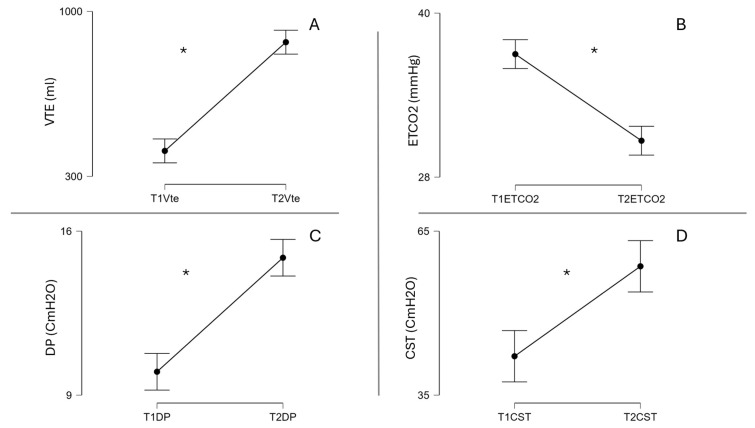
Values of VTE (**A**), ETCO_2_ (**B**), DP (**C**) and Cst (**D**) achieved in the EG during VHI. Values expressed as means and CIs. Cst: Static Compliance; VTE: Expired Tidal Volume; DP: driving pressure. * Statistical difference (*p* < 0.001)—*t*-test for paired samples.

**Figure 5 sensors-24-07066-f005:**
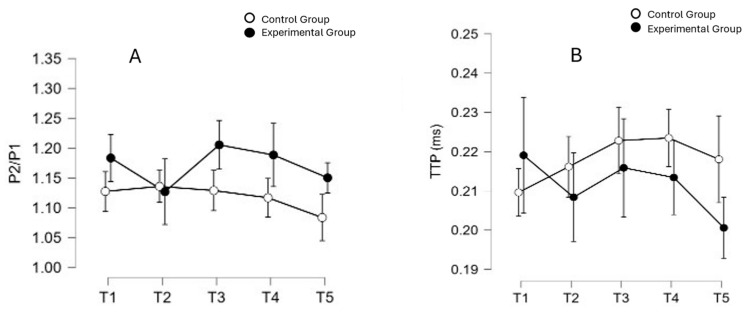
Mean values and confidence intervals for the P2/P1 ratio (**A**) and time to peak (TTP) (**B**) across five monitoring moments, categorized by groups. A two-way ANOVA was performed to evaluate differences among groups and moments and no statistical difference was found.

**Figure 6 sensors-24-07066-f006:**
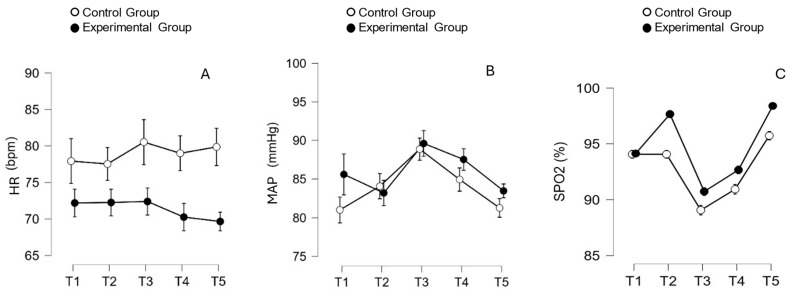
Mean values and confidence intervals for the heart rate (HR) (**A**), mean arterial pressure (MAP) (**B**) and peripheral oxygen saturation (SPO_2_) (**C**), measured at five monitoring times, categorized by groups. A two-way ANOVA was conducted to compare differences between groups and moments. Statistical differences between groups were found at T2 and T5 (Tukey’s post hoc *p* < 0.001).

**Table 1 sensors-24-07066-t001:** Baseline clinical characteristics.

	Total Sample (*n* = 30)	CG (*n* = 15)	EG (*n* = 15)	*p* Value
Age (years)	64.3 ± 13.9	66.4 ± 13.8	62.2 ± 14.1	0.411
Sex (M/F)	18/12	7/8	11/4	0.264
BMI (kg/m^2^)	21.6 ± 2.1	21.9 ± 2.0	21.2 ± 2.1	0.393
APACHE II	23.6 ± 3.7	22.2 ± 3.4	25 ± 3.6	0.045 *
RASS	−3.2 ± 0.6	−3.3 ± 0.7	−3.2 ± 0.5	0.577
Intravenous sedation (Y/N)	6/24	4/11	2/13	0.361
Baseline characteristics for respiratory variables				
Cst (L/cmH_2_O)	45.5 ± 12.7	48.9 ± 13.6	42.1 ± 11.0	0.142
Cdyn (L/cmH_2_O)	34.7 ± 8.78	37.7 ± 8.9	31.6 ± 7.7	0.054
R (cmH_2_O/L/s)	7 ± 3	6.1 ± 2.4	7.8 ± 3.3	0.123
SD (cmH_2_O)	9.43 ± 2.07	8.8 ± 2.1	10.0 ± 1.9	0.138
ETCO_2_ (mmHg)	37.5 ± 3.09	38.0 ± 3.6	37.0 ± 2.5	0.385
Severity scores				
P2/P1 Ratio	1.16 ± 0.176	1.127 ± 0.179	1.183 ± 0.176	0.395
TTP (ms)	0.21 ± 0.06	0.210 ± 0.054	0.219 ± 0.068	0.697
GCS < 8	30	15	15	
Type of brain injury				
HIC/HSA	23/7	11/4	12/3	1.000
ICH	23	11	12	
ICH score 3	10	3	7	
ICH score 4	5	4	1	
ICH score 5	8	4	4	
HSA	7	4	3	
FISHER II	3	1	2	
FISCHER III	3	2	1	
FISCHER IV	1	1	0	
Comorbidities				
1 Hypertension	20	12	8	
2. Diabetes	11	5	6	
3. Dyslipidaemia	4	2	2	
Dead in ICU (A/O)	20/10	11/4	9/6	0.700

Values are presented as mean ±SD or absolute frequency. M: Male; F: Female; BMI: Body Mass Index; APACHE II: Acute Physiology and Chronic Health Evaluation II; RASS: Richmond Agitation–Sedation Scale; Cst: Static Compliance; Cdyn: Dynamic Compliance; R: Resistance; DP drive pressure; ETCO_2_: End-Tidal Carbon Dioxide; TPP: Time To Pic; GCS: Glasgow Coma Scale; ICH: intracerebral hemorrhage; SAH: subarachnoid hemorrhage; A: ICU discharge; O: ICU death. * Statistical difference.

## Data Availability

The data supporting reported results can be made available upon reasonable request.

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
