# Peer review of "The Neurological and Hemodynamics Safety of an Airway Clearance Technique in Patients with Acute Brain Injury: An Analysis of Intracranial Pressure Pulse Morphology Using a Non-Invasive Sensor"

_sensors, 2024, doi:10.3390/s24217066_

Round 1
Reviewer 1 Report
Comments and Suggestions for Authors
Thank you for the opportunity to review this interesting manuscript. I believe it provides valuable information on the use of high tidal volume secretion mobilisation in neurocritical patients, and its effects on hemodynamic and neurological safety. I have only a few comments and questions.
- I think there is a little confusion as to whether this is physiological understanding paper or a sensor evaluation. The title and introduction focus somewhat on the novelty of the "new" sensor for assesment of ICP. However, it is only really used as a validated sensor, the study does not provide further evaluation. So this is a clinical/physiological study in my opinion, rather than a sensor study. I would suggest you review the introduction and see if you agree with me that the reader might be a little confused and perhaps chnage the tone with regard to the novelty of the sensor.
- I could not help wondering why you excluded hemodynamically unstable patients. I understand that these would likely have responded more to the tidal volume changes, but that would have been interesting and would have challenged the limits of the method. You have included this as a limitation in the discussion, so there is not anything to do.
- In a similar way, these patients do not seem to have very sever pulmonary abnormalities. It seems you are able to obtain tidal volumes increasing to 800 ml with only a driving pressure increase from 9-16 cmH2O. I realise that for the majority of neurological patients, this level of pulmomary disfunction is probably common, so the results are typical. You may wish however to provide a point in the discussion on the potential limitation in more severe lung disease, in a similar way to hemodynamic status
In table 1 you write SG, I presume you mean EG.
Author Response
1) I think there is a little confusion as to whether this is physiological understanding
paper or a sensor evaluation. The title and introduction focus somewhat on the novelty of
the "new" sensor for assesment of ICP. However, it is only really used as a validated
sensor, the study does not provide further evaluation. So this is a clinical/physiological
study in my opinion, rather than a sensor study. I would suggest you review the
introduction and see if you agree with me that the reader might be a little confused and
perhaps chanage the tone with regard to the novelty of the sensor.
Response: Thank you for your observation. We agree with you, and the introduction was
revised and changed.
2) I could not help wondering why you excluded hemodynamically unstable patients.
I understand that these would likely have responded more to the tidal volume changes,
but that would have been interesting and would have challenged the limits of the method.
You have included this as a limitation in the discussion, so there is not anything to do.
Response: Thank you for your comment. With regard to this point, it is important to
mention that there are still no studies in the literature that have investigated the influence
of a maneuver to remove pulmonary secretions by increasing positive pressure, and its
influence on hemodynamics in patients with acute brain injury, this being the first point of
investigation for this group. It is also important to mention that patients with acute brain
injury have strict control of mean arterial pressure as a way of maintaining cerebral
perfusion pressure. Thus, subjecting unstable patients to a therapy that tends to cause
hypotension due to increased intrathoracic pressure could lead to secondary brain
damage due to decreased cerebral perfusion pressure, which justifies initially
investigating only hemodynamically stable patients. We have included a brief discussion
of this point in the limitations of this study.
3) In a similar way, these patients do not seem to have very sever pulmonary
abnormalities. It seems you are able to obtain tidal volumes increasing to 800 ml with only
a driving pressure increase from 9-16 cmH2O. I realise that for the majority of neurological
patients, this level of pulmomary disfunction is probably common, so the results are
typical. You may wish however to provide a point in the discussion on the potential
limitation in more severe lung disease, in a similar way to hemodynamic status
Response: Thank you for your comment. Similarly to the previous item, this group chose
to initially investigate the effects of a bronchial secretion removal maneuver in patients
with acute brain injury and no lung injury, for a few reasons: there are no studies of this
patient profile in the literature, this is the most common patient profile in clinical practice
and because there are studies that recommend different mechanical ventilation
parameters in the case of acute brain injury patients with lung injury (which would require
a differently designed protocol with greater ventilatory, hemodynamic and neurological
control). We have included a brief discussion of this point in the limitations of this study.
4) In table 1 you write SG, I presume you mean EG.
Response: Thank you for your comment. The terms in table 1 and throughout the text
have been changed.

Reviewer 2 Report
Comments and Suggestions for Authors
Souza et al evaluated effects on intracranial pressure and hemodynamics in patients with acute brain injury who received bronchial secretion removal technique devices. It is critical to assess neurological safety for patients under mechanical ventilation, especially in patients with acute brain injury whose vital signs are counting on both cerebral and pulmonary functions. The authors utilized non-invasive sensors and recently developed devices, which could help the field improve current therapeutic strategies for patients. However, in my opinion, the manuscript still exhibits some major and minor issues listed as follows
1. Parameters used to assess cerebral functions are descriptive measurements for morphology of the ICP curve, which only infers one aspect of mechanical properties of the brain. Therefore, results from the manuscript cannot extend to cover all aspects of ‘neurological safety’, which has been used many times in the manuscript. Please be specific for all of them.
2. Manuscripts especially figures are poorly prepared. Figures 3 and 4 lack proper labels for the axes. Figure 5 was missing. Citations of figures were missed in the main text. Descriptive for statistical analyses should be added to the figure legends instead of being mentioned in the main text. Figure 6 should include a comparison between the two groups instead of just showing measurements at different time points within the same treatment group.
Comments on the Quality of English Language
English in general is okay
Author Response
1. Parameters used to assess cerebral functions are descriptive measurements for
morphology of the ICP curve, which only infers one aspect of mechanical
properties of the brain. Therefore, results from the manuscript cannot extend to
cover all aspects of ‘neurological safety’, which has been used many times in the
manuscript. Please be specific for all of them.
Response: Thank you for your observation. Thank you for your comment. We agree with
you that the aspect evaluated by this work refers to mechanical properties. However, as
mentioned in the title of the paper and throughout it, we clarify that the aspect evaluated
is the wave morphology of intracranial pressure, by a sensor validated against the current
gold standard, capable of assessing the risk of intracranial hypertension and intracranial
compliance. We have supplemented this information throughout the text, and put this
point forward as a limitation of the study, as well as an interesting point to be analyzed
and added to in future research.
2. Manuscripts especially figures are poorly prepared. Figures 3 and 4 lack proper
labels for the axes. Figure 5 was missing. Citations of figures were missed in the
main text. Descriptive for statistical analyses should be added to the figure legends
instead of being mentioned in the main text. Figure 6 should include a comparison
between the two groups instead of just showing measurements at different time
points within the same treatment group.
Response: Thank you for your comment. The recommended changes have been made.
Figure 3 cannot be modified as it is an image taken from the B4C report, for further
clarification we have added information in the caption.

Round 2
Reviewer 2 Report
Comments and Suggestions for Authors
Statistics are still not clearly stated in the figure legends such as significant effects in Figure 6. The language used is not accurate either. Please carefully check your manuscript and polish the wording.
Comments on the Quality of English LanguageEnglish needs to be carefully checked and edited.
Author Response
- Statistics are still not clearly stated in the figure legends such as significant effects in Figure 6. The language used is not accurate either. Please carefully check your manuscript and polish the wording.
Response: Answer: Thank you for your comment. The suggested changes have been made.
As requested, the English has been carefully checked and edited by an expert in the language. As a result, the title and abstract have also been modified.
